artificial intelligence/pattern recognition/ mechanical engineering

rolling element bearing, pattern recognition, Holder coefficient characteristics, fractal box-counting dimension characteristics, grey relation algorithm, Yager algorithm

**Author for correspondence:**
Yulong Ying
e-mail: yingyulong060313@163.com

# Research on rolling bearing fault diagnosis based on multi-dimensional feature extraction and evidence fusion theory

Jingchao Li[1], Yulong Ying[2], Yuan Ren[1], Siyu Xu[2], Dongyuan Bi[1], Xiaoyun Chen[1] and Yufang Xu[1]

[1]College of Electronic and Information Engineering, Shanghai Dianji University, Shanghai, People's Republic of China
[2]School of Energy and Mechanical Engineering, Shanghai University of Electric Power, Shanghai 200090, People's Republic of China

YY, 0000-0002-3867-5893

Rolling bearing failure is the main cause of failure of rotating machinery, and leads to huge economic losses. The demand of the technique on rolling bearing fault diagnosis in industrial applications is increasing. With the development of artificial intelligence, the procedure of rolling bearing fault diagnosis is more and more treated as a procedure of pattern recognition, and its effectiveness and reliability mainly depend on the selection of dominant characteristic vector of the fault features. In this paper, a novel diagnostic framework for rolling bearing faults based on multi-dimensional feature extraction and evidence fusion theory is proposed to fulfil the requirements for effective assessment of different fault types and severities with real-time computational performance. Firstly, a multi-dimensional feature extraction strategy on the basis of entropy characteristics, Holder coefficient characteristics and improved generalized box-counting dimension characteristics is executed for extracting health status feature vectors from vibration signals. And, secondly, a grey relation algorithm is used to calculate the basic belief assignments (BBAs) using the extracted feature vectors, and lastly, the BBAs are fused through the Yager algorithm for achieving bearing fault pattern recognition. The related experimental study has illustrated the proposed method can effectively and efficiently recognize various fault types and severities in comparison with the existing intelligent diagnostic methods based on a small number of training samples with good real-time performance, and may be used for online assessment.

# 1. Introduction

The rolling bearing as an important part is widely used in almost all types of rotating machinery, such as gas turbines, steam turbines and diesel engines. Rolling bearing failure is one of the main causes of failure and damage for rotating machinery, and leads to huge economic losses [1–3]. To ensure reliable operation of the rotating machinery and reduce economic losses, it is necessary to propose a reliable and effective fault diagnosis method for the rolling bearing. Among many fault diagnosis approaches for rolling bearings, vibration-based diagnostic methods have received much attention in the past few decades [4,5]. Bearing vibration signals contain a wealth of information on mechanical health status, which also makes it possible to extract the dominant features that characterize the mechanical health status from vibration signals through signal processing techniques [6]. Currently, many signal processing techniques have been applied to bearing offline fault diagnosis. However, due to many nonlinear factors (e.g. stiffness, friction, clearance, etc.), bearing vibration signals (especially in a faulted condition) will exhibit nonlinear and unsteady character [7]. In addition, the measured vibration signal contains not only information about the operating conditions associated with bearing itself, but also information on a large number of other rotating components and structures in the plant equipment [8]. Owing to large background noise, slight bearing fault information is easily submerged in the background noise. Therefore, conventional time-domain and frequency-domain methods may not easily make an accurate assessment of the bearing health status [9]. With the development of nonlinear dynamics, many nonlinear analytical techniques have been applied to identify and predict the complex dynamic nonlinearities for rolling bearings [10]. Among them, the most typical one is to extract the fault signature frequency from vibration signals through the combined usage of some advanced signal processing techniques (such as HOS [11], WPT [12], Hilbert transform [13], empirical mode decomposition (EMD), etc.) and further evaluate the bearing health status by comparing extracted fault signature frequency with the theoretical characteristic frequency value with involvement of expert empirical judgement.

With the development of artificial intelligence [14], bearing fault diagnosis is more and more treated as the category of pattern recognition. And its effectiveness and reliability mainly depend on the selection of dominant eigenvectors that characterize the fault features [15]. Recently, some entropy-based methods (such as hierarchical entropy [16], fuzzy entropy [17], sample entropy (SampEn) [18], approximate entropy [19,20], hierarchical fuzzy entropy, etc.) have been proposed for extracting dominant eigenvectors from bearing vibration signals and have achieved some effect. In this paper, a novel diagnostic framework for rolling bearing faults based on multi-dimensional feature extraction and evidence fusion theory is proposed and multi-dimensional feature extraction on the basis of entropy characteristics, Holder coefficient characteristics and improved generalized box-counting dimension characteristics is performed for extracting health status feature vectors from bearing vibration signals.

After fault feature extraction, a pattern recognition technique is required to achieve the rolling element bearing fault diagnosis automatically [15]. Nowadays, a variety of pattern recognition methods have been used in mechanical fault diagnosis, of which the most widely used are the support vector machines (SVMs) [21] and artificial neural networks (ANNs) [22–24]. Among them, the ANN training requires a large number of samples, which is difficult or even impossible to achieve in practical applications, especially the samples with fault features. The SVMs are based on statistical learning theory (especially suitable for training small samples), which have better generalization ability than ANNs and can ensure that the local optimal solutions and global optimal solutions are exactly consistent [25]. However, the accuracy of SVM classifiers depends on the selection of their optimal parameters [25,26]. In order to ensure the diagnostic accuracy, some optimization algorithms and/or the design of complex multi-class structures [27] often need to be used complementally to improve the effectiveness of SVMs. In this paper, in order to solve the issue of generality versus accuracy, a grey relation algorithm (GRA) is used to calculate the basic belief assignments (BBAs) using the extracted feature vectors based on multi-dimensional feature extraction, and the BBAs are fused through the Yager algorithm for achieving bearing fault pattern recognition based on a small number of training samples.

In summary, this paper aims to solve the problem that the traditional time and frequency-domain methods are not easy for making an accurate assessment of the health status of rolling bearings, and a novel diagnostic framework is proposed. The rest of the paper is organized as follows. Firstly, the diagnostic framework of the proposed method is introduced in §2, and secondly, the related experimental study of the proposed method is illustrated in §3 and the conclusion is presented in §4.

# 2. Methodology

## 2.1. Multi-dimensional feature extraction

In this paper, a novel diagnostic framework for rolling bearing faults based on multi-dimensional feature extraction and evidence fusion theory was developed to meet the requirements for accurate assessment of different fault types and severities with real-time computational performance. Firstly, multi-dimensional feature extraction on the basis of entropy characteristics, Holder coefficient characteristics and improved generalized box-counting dimension characteristics were proposed for extracting health status feature vectors from bearing vibration signals, respectively.

### 2.1.1. Entropy characteristics

Entropy is a crucial concept in information theory and is a measure for information uncertainty of signal distribution and a measure for signal complexity [28]. Therefore, the information contained within signals can be quantitatively described by entropy characteristics.

Suppose the bearing vibration signal is $f$. The signal $f$ is sampled and discretized into a discrete signal sequence $f(i)$, $i = 1, 2, \ldots, n$, where $n$ is the total number of the discrete signal points. Perform fast Fourier transform (FFT) as follows:

$$F(k) = \sum_{i=0}^{n-1} f(i) \exp\left(-j\frac{2\pi}{n}ik\right) \quad k = 0, 1, \ldots, n - 1. \tag{2.1}$$

After obtaining the signal spectrum, calculate the energy of each point:

$$E_k = |F(k)|^2. \tag{2.2}$$

Calculate the total energy of the signal spectrum:

$$E = \sum_{k=0}^{n-1} E_k. \tag{2.3}$$

Calculate the ratio of the energy of each point to the total energy of the signal spectrum:

$$P_k = \frac{E_k}{E} = \frac{E_k}{\sum_{k=0}^{n-1} E_k}. \tag{2.4}$$

The Shannon entropy $E_1$ and exponential entropy $E_2$ can be defined as follows:

$$E_1 = -\sum_{k=0}^{n-1} P_k \log P_k \tag{2.5}$$

and

$$E_2 = \sum_{k=0}^{n-1} P_k e^{1-P_k}. \tag{2.6}$$

The entropy characteristics $[E_1, E_2]$ are taken as a part of dominant feature vectors for rolling element bearing fault pattern recognition.

### 2.1.2. Holder coefficient characteristics

The Holder coefficient algorithm evolves from the Holder inequality [29,30]. The Holder coefficient can be used to measure similar degree of two discrete signal sequences. The definition of the Holder inequality can be described as follows:

For any vector $X = [x_1, x_2, \ldots, x_n]^T$ and $Y = [y_1, y_2, \ldots, y_n]^T$, they satisfy:

$$\sum_{i=1}^{n} |x_i \cdot y_i| \leq \left(\sum_{i=1}^{n} |x_i|^p\right)^{1/p} \cdot \left(\sum_{i=1}^{n} |y_i|^q\right)^{1/q}, \tag{2.7}$$

where $1/p + 1/q = 1$ and $p, q > 1$.

Based on the Holder inequality, for two discrete signal sequences $\{f_1(i) \geq 0, \ i = 1, 2, \ldots, n\}$ and $\{f_2(i) \geq 0, \ i = 1, \ 2, \ldots, n\}$, if $1/p + 1/q = 1$ and $p, \ q > 1$, then the Holder coefficient of these two discrete signal sequences is obtained as follows:

$$H_c = \frac{\sum_{i=1}^{n} f_1(i)f_2(i)}{\left(\sum_{i=1}^{n} f_1^p(i)\right)^{1/p} \cdot \left(\sum_{i=1}^{n} f_2^q(i)\right)^{1/q}}, \tag{2.8}$$

where $0 \leq H_c \leq 1$.

The Holder coefficient characterizes similar degree of two discrete signal sequences, and if and only if $f_1^p(i) = k f_2^q(i), i = 1, 2, \ldots, n$, in which $n$ denotes the length of the discrete signal sequence and $k$ is a real number, $H_c$ will be the maximum value. In this case, the similar degree of two discrete signal sequences is biggest, which indicates these two discrete signal sequences belong to the same type; if and only if $\sum_{i=1}^{n} f_1(i)f_2(i) = 0$, $H_c$ will be the minimum value, and in this case, the similar degree of the two discrete signal sequences is smallest, which indicates these two signal sequences are irrelevant and belong to different types.

The rectangular signal sequence $s_1(i)$ and the triangular signal sequence $s_2(i)$ are selected as two reference sequences in the paper. Calculate the Holder coefficient values between the bearing vibration signal sequence $f(i)$ and the two reference signal sequences, respectively.

Calculate the Holder coefficient value $H_1$ between the bearing vibration signal sequence $f(i)$ and the rectangular signal sequence $s_1(i)$.

$$H_1 = \frac{\sum_{i=1}^{n} f(i)s_1(i)}{\left(\sum_{i=1}^{n} f^p(i)\right)^{1/p} \cdot \left(\sum_{i=1}^{n} s_1^q(i)\right)^{1/q}}, \tag{2.9}$$

where the rectangular signal sequence $s_1(i)$ is expressed as:

$$s_1(i) = \begin{cases} 1, & 1 \leq i \leq n \\ 0, & \text{else} \end{cases}. \tag{2.10}$$

In the same way, we obtain the Holder coefficient value $H_2$ between the vibration signal sequence $f(i)$ and the triangular signal sequence $s_2(i)$.

$$H_2 = \frac{\sum_{i=1}^{n} f(i)s_2(i)}{\left(\sum_{i=1}^{n} f^p(i)\right)^{1/p} \cdot \left(\sum_{i=1}^{n} s_2^q(i)\right)^{1/q}}, \tag{2.11}$$

where the triangular signal sequence $s_2(i)$ is expressed as:

$$s_2(i) = \begin{cases} \dfrac{2i}{n}, & 1 \leq i \leq \dfrac{n}{2} \\ \dfrac{2 - 2i}{n}, & \dfrac{n}{2} \leq i \leq n \end{cases}. \tag{2.12}$$

The Holder coefficient characteristics $[H_1, H_2]$ are taken as a part of dominant feature vectors for rolling element bearing fault pattern recognition.

### 2.1.3. Fractal box-counting dimension characteristics

Fractal theory is one of the most important branches for contemporary nonlinear science, and is suitable for processing all types of nonlinear and non-stationary phenomenon and may also be suitable for fault feature extraction from bearing vibration signals. Fractal box-counting dimension algorithm has the advantage of simple calculation compared with other fractal dimension algorithms. The conventional algorithm of fractal box-counting dimension has been widely used in the fields of image analysis, electromagnetic fault diagnosis and biomedicine, which have strict self-similar signals.

Suppose $A$ is a non-empty bounded subset of Euclidean space $R^n$ to be calculated, and $N(A, \varepsilon)$ is the least number of boxes with the side length $\varepsilon$ covering $A$. Then the box-counting dimension can be defined as:

$$D = \lim_{\varepsilon \to 0} \frac{\log N(A, \varepsilon)}{\log (1/\varepsilon)}. \tag{2.13}$$

For the actual sampled vibration signal sequence $f(i), i = 1, 2, \ldots, N_0$, there is no meaning for $\varepsilon \to 0$ to calculate the box-counting dimension as the sampling interval $\sigma$ is the highest resolution for the signal

due to the existence of sampling frequency. It is often made the minimum side length of the box $\varepsilon = \sigma$. Consider the actual sampled bearing vibration signal sequence $f(i)$ as the closed set of Euclidean space $R^n$, and the calculation process of box-counting dimension is described as follows.

Use the approximate method to make the minimum side length of the box covering the vibration discrete signal sequence $f(i)$ equal to the sampling interval $\sigma$. And calculate the least number of boxes $N(k\varepsilon)$ with side length $k\varepsilon$ covering the signal sequence $f(i)$, thus:

$$p_1 = \max\{f(k(i-1)+1), f(k(i-1)+2), \ldots, f(k(i-1)+k+1)\}, \tag{2.14}$$

$$p_2 = \min\{f(k(i-1)+1), f(k(i-1)+2), \ldots, f(k(i-1)+k+1)\} \tag{2.15}$$

and

$$p(k\varepsilon) = \sum_{i=1}^{N_0/k} |p_1 - p_2|, \tag{2.16}$$

where $i = 1, 2, \ldots, N_0/k$, $k = 1, 2, \ldots, K$. $N_0$ is the number of sampling points, $K < N_0$. $p(k\varepsilon)$ is the longitudinal coordinate scale of the actual sampled bearing vibration signal sequence $f(i)$. Thus, $N(k\varepsilon)$ can be defined as:

$$N(k\varepsilon) = \frac{p(k\varepsilon)}{k\varepsilon + 1}. \tag{2.17}$$

Select a fitting curve $\log k\varepsilon \sim \log N(k\varepsilon)$ with good linearity as a scale-free zone, and the fitting curve can be defined as:

$$\log N(k\varepsilon) = a \log k\varepsilon + b, \tag{2.18}$$

where $k_1 \leq k \leq k_2$ and $k_1$ and $k_2$ are the start and end of the scale-free zone, respectively.

Generally, a least square method is used to calculate the slope of the fitting curve, which is the fractal box-counting dimension $D$ of the actual sampled bearing vibration signal sequence $f(i)$:

$$D = -\frac{(k_2 - k_1 + 1)\sum \log k \cdot \log N(k\varepsilon) - \sum \log k \cdot \sum \log N(k\varepsilon)}{(k_2 - k_1 + 1)\sum \log^2 k - (\sum \log k)^2}. \tag{2.19}$$

However, for the actual bearing vibration signals, they do not satisfy the self-similar structure of fractal theory to some degree. Therefore, when using the traditional fractal box-counting dimension algorithm to calculate box-counting dimension of the vibration signals, the fitting curve often does not have good linear structure. Aiming at this issue, an improved generalized fractal box-counting dimension algorithm has been developed to overcome the defect of the conventional fractal box-counting dimension algorithm. The specific calculation procedure is as follows:

(i) Resample the actual bearing vibration signal sequence $f(i)$, $i = 1, 2, \ldots, N_0$, and properly increase the sampling points to reduce the minimum side length $\varepsilon$, to improve the calculation accuracy of the fractal box-counting dimension of the signal sequence $f(i)$. The phase space of the signal sequence $f(i)$ is reconstructed, and the number of iterated dimension of the reconstructed phase space is determined according to the number of sampling points.

(ii) Suppose the number of sampling points of the signal sequence $f(i)$ is $N_0 = 2^n$. To improve the calculation accuracy, resample the actual bearing vibration signal sequence $f(i)$, and suppose the number of sampling points of the signal sequence $f(i)$ is $N = 2^K (K > n)$. The reconstruction dimension of the phase space of the signal sequence $f(i)$ is set, respectively, as $m = K + 1 = 2, 3, 4, \ldots, \log_2 N + 1$.

(iii) The derived process of the number of boxes covering the actual bearing vibration signal sequence $f(i)$ can be described as follows:

when $k = 1$:

$p_1 = \max\{f(i), f(i+1)\}, p_2 = \min\{f(i), f(i+1)\}, i = 1, 2, \ldots, N/k$. In this case, the reconstructed phase space dimension is 2;

when $k = 2$:

$p_1 = \max\{f(2i-1), f(2i), f(2i+1)\}, p_2 = \min\{f(2i-1), f(2i), f(2i+1)\}, i = 1, 2, \ldots, N/k$. In this case, the reconstructed phase space dimension is 3;

when $k = 3$:

$p_1 = \max\{f(3i-2), f(3i-1), f(3i), f(3i+1)\}, p_2 = \min\{f(3i-2), f(3i-1), f(3i), f(3i+1)\}, i = 1, 2, \ldots, N/k$. In this case, the reconstructed phase space dimension is 4;

when $k = K$:

$p_1 = \max\{f(Ki - K + 1), f(Ki - K + 2), \ldots, f(Ki + 1)\}, p_2 = \min\{f(Ki - K + 1), f(Ki - K + 2), \ldots, f(Ki + 1)\}, i = 1, 2, \ldots, N/k$. In this case, the reconstructed phase space dimension is $m = K + 1$.

(iv) It can be seen from the above deduction that, during reconstructing the phase space of the bearing vibration signal sequence $f(i)$ $K$ times, the corresponding $\log N(k\varepsilon)$ can be obtained at each time. And then the relationship curve of $\log k\varepsilon \sim \log N(k\varepsilon)$ can be determined. Since the fitting curve does not have a strict linear relationship, calculate the derivative of the relationship curve at these $K$ points. The slopes $D_1, D_2, D_3, \ldots, D_K$ at these $K$ points from the relationship curve are the fractal box-counting dimensions in the different reconstructed phase space. Take the slopes $D_1, D_2, D_3, \ldots, D_K$ obtained as the $K$ characteristic parameters for the fault feature vector extracted from the signal sequence $f(i)$, which characterizes the bearing fault symptoms.

## 2.2. Basic belief assignment acquisition

The research of grey relation theory is the foundation of grey system theory, which is mainly based on the basic theory of space mathematics to calculate relation coefficient and relation degree between reference characteristic vector and each comparative characteristic vector. GRA has good potential to be used in BBA acquisition for rolling element bearing fault pattern recognition with the following reasons [31,32]: it has good tolerance to measurement noise; its algorithm is simple and can solve the issue of generality versus accuracy; and it can solve the learning problem with a small number of samples.

Suppose the health status feature vectors, i.e. the multi-dimensional feature vectors extracted based on entropy characteristics, the Holder coefficient characteristics and improved generalized box-counting dimension characteristics, from vibration signals, to be recognized, are as follows:

$$B_1 = \begin{bmatrix} b_1(1) \\ b_1(2) \\ b_1(3) \\ \ldots \\ b_1(q) \end{bmatrix}, \quad B_2 = \begin{bmatrix} b_2(1) \\ b_2(2) \\ b_2(3) \\ \ldots \\ b_2(q) \end{bmatrix}, \quad B_i = \begin{bmatrix} b_i(1) \\ b_i(2) \\ b_i(3) \\ \ldots \\ b_i(q) \end{bmatrix}, \tag{2.20}$$

where $B_i$ $(i = 1, 2, \ldots)$ is a certain fault pattern to be recognized (i.e. fault types and severities). $q$ is the total number of characteristic parameters chosen as the characteristic vector, where $q = 2$ for entropy characteristics $[E_1, E_2]^T$ and Holder coefficient characteristics $[H_1, H_2]^T$, and $q = K$ for improved generalized fractal box-counting dimension characteristics $[D_1, D_2, D_3, \ldots, D_K]^T$, respectively.

Assume that the knowledge base (i.e. the recognition template) between the fault patterns (i.e. fault types and severities) and fault signatures (i.e. the fault feature vectors) based on a small number of training samples is as follows:

$$C_1 = \begin{bmatrix} c_1(1) \\ c_1(2) \\ c_1(3) \\ \ldots \\ c_1(q) \end{bmatrix}, \quad C_2 = \begin{bmatrix} c_2(1) \\ c_2(2) \\ c_2(3) \\ \ldots \\ c_2(q) \end{bmatrix}, \quad C_j = \begin{bmatrix} c_j(1) \\ c_j(2) \\ c_j(3) \\ \ldots \\ c_j(q) \end{bmatrix}, \tag{2.21}$$

where $C_j$ $(j = 1, 2, \ldots)$ is a known fault pattern and $c_j$ $(j = 1, 2, \ldots)$ is a certain feature parameter.

$$\text{For } \rho \in (0, 1): \xi(b_i(k), c_j(k)) = \frac{\displaystyle\min_j \min_k |b_i(k) - c_j(k)| + \rho \cdot \max_j \max_k |b_i(k) - c_j(k)|}{|b_i(k) - c_j(k)| + \rho \cdot \max_j \max_k |b_i(k) - c_j(k)|} \tag{2.22}$$

and

$$\xi(B_i, C_j) = \frac{1}{q} \sum_{k=1}^{q} \xi(b_i(k), c_j(k)), j = 1, 2, \ldots \tag{2.23}$$

where $\rho$ is the distinguishing coefficient; $\xi(b_i(k), c_j(k))$ is the grey relation coefficient of $k$th characteristic parameter for $B_i$ and $C_j$; and $\xi(B_i, C_j)$ is the grey relation degree between $B_i$ and $C_j$.

According to the matching degree between the feature vectors to be recognized and the recognition template by the GRA, the basic probability assignment function for BBA can be obtained.

## 2.3. Basic belief assignments fusion

In this step, the BBAs obtained by GRA are fused through the Yager algorithm for achieving bearing fault pattern recognition intelligently using the extracted feature vectors.

First of all, the Dempster–Shafer evidence theory is introduced as the following. Let $\Theta$ be the universe, and the set $\Theta$ representing all possible states of a system under consideration. The power set

$2^{\Theta}$ is the set of all subsets of $\Theta$, including the empty set $\phi$. For example, if: $\Theta = \{F_1, F_2\}$, then $2^{\Theta} = \{\phi, \{F_1\}, \{F_2\}, \Theta\}$. The elements of the power set can be taken to represent propositions concerning the actual state of the system, by containing all and only the states in which the proposition is true. The theory of evidence assigns a belief mass to each element of the power set. Formally, a function: $m : 2^{\Theta} \rightarrow [0, 1]$ is called a BBA, when it has two properties. The mass of the empty set is zero, that is, $m(\phi) = 0$, and the masses of the remaining members of the power set add up to a total of 1, that is $\sum_{A \in 2^{\Theta}} m(A) = 1$.

The mass $m(A)$ of $A$, a given member of the power set, represents the proportion of all relevant and available evidence which supports the claim that the actual state belongs to $A$ but to no particular subset of $A$. The value of $m(A)$ pertains only to the set $A$ and makes no additional claims about any subsets of $A$.

In particular, the combination (called the joint mass) is calculated from the two masses $m_1$ and $m_2$ in the following manner:

$$m_{1,2}(\phi) = 0, \tag{2.24}$$

$$m_{1,2}(A) = (m_1 \oplus m_2)(A) = \frac{1}{1-k} \sum_{B \cap C = A \neq \phi} m_1(B) m_2(C) \tag{2.25}$$

$$\text{and} \quad k = \sum_{B \cap C = \phi} m_1(B) m_2(C), \tag{2.26}$$

where $k$ is a measure of the conflicting amount between the two mass sets.

The Dempster–Shafer evidence theory is a critical method to fuse the results from multi-symptom domain. However, when dealing with highly conflicted evidence, the Dempster–Shafer evidence theory will lead to an abnormal conclusion. Aiming at this problem, some researchers presented many improved combination rules, for example, the Yager method, Dubois and Prade method and Smets method. In this paper, the improved combination rule of the Yager method was used to fuse all the BBAs to obtain the last decision-making results. The improvement of the Yager method is to use the conflicting coefficient $k$ as an evidence of uncertainty and a new mathematical model can be obtained as follows.

$$m_{1,2}(A) = (m_1 \oplus m_2)(A) = \begin{cases} \sum_{B \cap C = A} m_1(B) m_2(C) & A \subset \Theta, A \neq \Theta \\ \sum_{B \cap C = \Theta} m_1(B) m_2(C) + k & A = \Theta \end{cases} . \tag{2.27}$$

## 2.4. Diagnostic procedure

Totally, the process of the proposed diagnostic framework (illustrated in figure 1) for rolling bearing fault diagnosis is as follows.

Step 1: The vibration signals from the object bearing are sampled under different health status, including normal operating condition and conditions with different fault types and severities, to establish the knowledge base (i.e. the recognition template).

Step 2: The health status feature vectors are extracted from the sample knowledge base through the multi-dimensional feature extraction based on entropy characteristics $[E_1, E_2]$, Holder coefficient characteristics $[H_1, H_2]$ and improved generalized box-counting dimension characteristics $[D_1, D_2, D_3, \ldots, D_K]$, respectively.

Step 3: The sample knowledge base is established based on the fault symptom (i.e. the extracted fault feature vectors) and the fault pattern (i.e. the known fault types and severities).

Step 4: The health status feature vectors extracted based on bearing vibration signals to be identified are input into the GRA to obtain BBAs (i.e. BBA1, BBA2, BBA3), and then the BBAs are fused through the Yager method to output the diagnostic results (i.e. fault types and severities).

# 3. Experimental validation

In this paper, the rolling bearing vibration signals for testing are from Case Western Reserve University Bearing Data Center [33]. The related rolling element bearing experimental device consists of a torque meter, a power meter and a three-phase induction motor, and the load power and speed are measured by the sensors, as shown in figure 2. The motor drive end rotor is supported by a test bearing, where a single point of failure is set by means of discharge machining. The test bearing is a deep groove rolling bearing of 6205-2RS JEM SKF. Through controlling the power meter, the desired

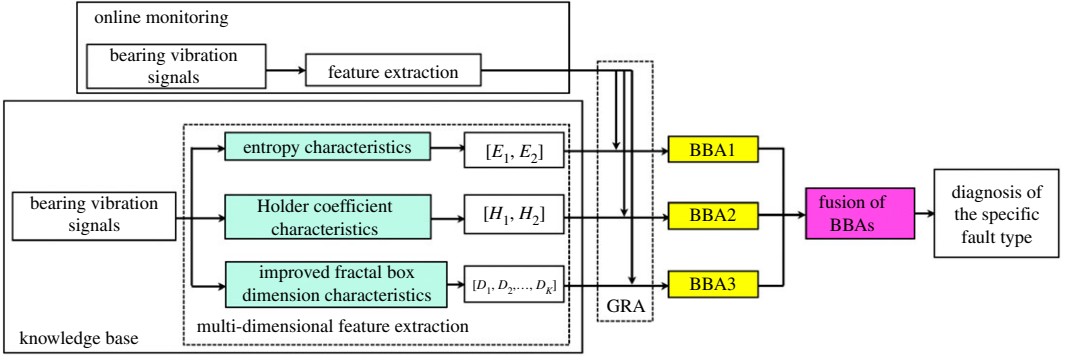

**Figure 1.** The diagnostic framework for rolling bearing fault diagnosis.

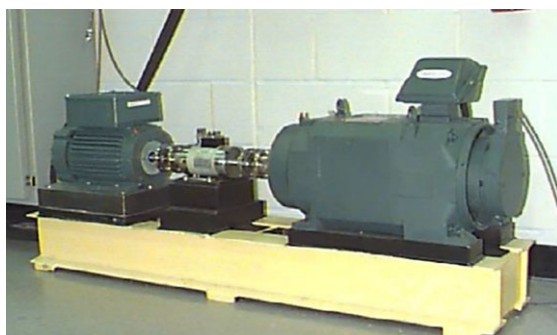

**Figure 2.** Experimental set-up.

torque load can be obtained. The fault types contain outer race fault, the inner race fault, and the ball fault, and the fault diameters, i.e. fault severities, contain 28, 21, 14 and 7‰. An accelerometer is installed on the motor drive end housing with a bandwidth of up to 5000 Hz, and the vibration data for the test bearing under different fault patterns are collected by a recorder, in which the sampling frequency is 12 kHz.

The bearing vibration data used for analysis are obtained under the motor speed of 1797 r min$^{-1}$ and load of 0 hp. Totally, 11 types of vibration signals considering different fault categories and severities are analysed, as seen in table 1. Each data sample from vibration signals is made up of 2048 time-series points. For those 550 data samples, 110 data samples are chosen randomly for the establishment of the knowledge base, with the rest of the 440 data samples taken as testing data samples. If the motor speed and the load change in the practical application, data samples under these working conditions have to be chosen for the establishment of the knowledge base so that the motor speed and load would not affect the diagnostic performance.

The health status feature vectors extracted from rolling bearing normal operating condition and different fault conditions with 7‰ fault diameter (figure 3) based on entropy characteristics, Holder coefficient characteristics and improved generalized box-counting dimension characteristics, are shown in figures 4–6, respectively. And the health status feature vectors extracted from inner race fault condition with various severities (figure 7) based on entropy characteristics, Holder coefficient characteristics and improved generalized box-counting dimension characteristics, are shown in figures 8–10, respectively.

From figures 4–6, the Holder coefficient characteristics show a better inter-class separation and intra-class polymerization than entropy characteristics and improved generalized box-counting dimension characteristics extracted from the bearing vibration signals with different fault types. However, from figures 8–10, the improved generalized box-counting dimension characteristics show a better inter-class separation than entropy characteristics and Holder coefficient characteristics extracted from the bearing vibration signals with different severities. As entropy characteristics, Holder coefficient characteristics and improved generalized box-counting dimension characteristics show their strengths and weaknesses in classifying different fault types and severities, evidence fusion theory is used to obtain the final diagnostic results. The knowledge base (i.e. the recognition template) for GRA is established based on the fault symptom (i.e. the

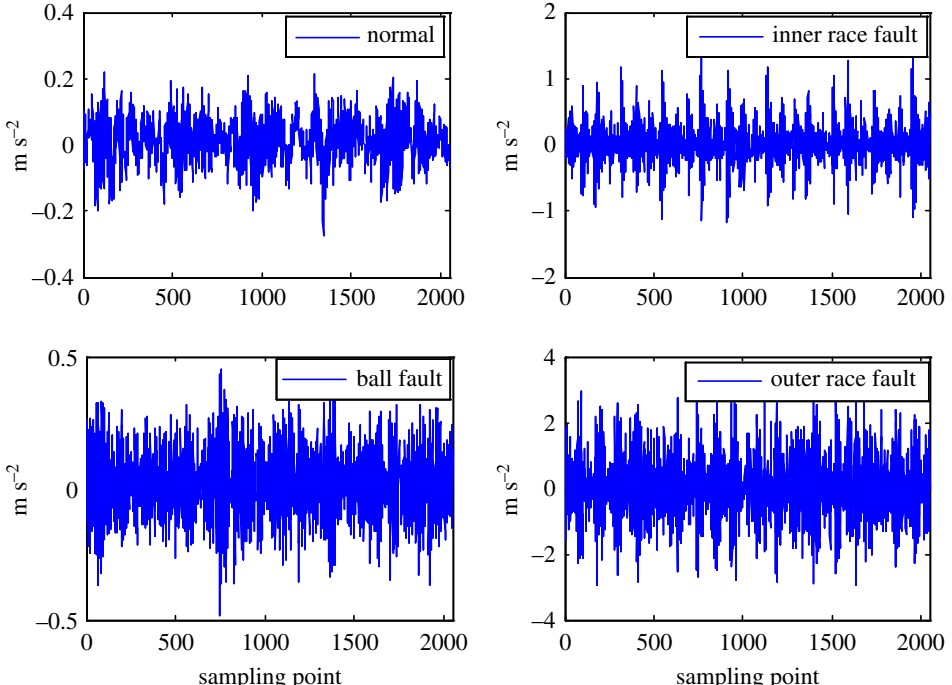

**Figure 3.** Rolling bearing normal operating condition and various fault conditions with fault diameter 7‰.

**Table 1.** Description of experimental dataset.

| health status condition | fault diameter (‰) | the number of base samples | the number of testing samples | label of classification |
|---|---|---|---|---|
| normal | 0 | 10 | 40 | 1 |
| inner race fault | 7 | 10 | 40 | 2 |
| | 14 | 10 | 40 | 3 |
| | 21 | 10 | 40 | 4 |
| | 28 | 10 | 40 | 5 |
| ball fault | 7 | 10 | 40 | 6 |
| | 14 | 10 | 40 | 7 |
| | 28 | 10 | 40 | 8 |
| outer race fault | 7 | 10 | 40 | 9 |
| | 14 | 10 | 40 | 10 |
| | 21 | 10 | 40 | 11 |

extracted feature vectors) and the fault pattern (i.e. the known fault types and severities). The fault feature vectors extracted based on the testing rolling bearing vibration signals to be recognized input to GRA, and the diagnostic results (i.e. fault types and severities) are output after the fusion of BBAs, shown in table 2.

The diagnostic results from table 2 show that the detecting success rate for bearing faulty conditions can reach 100%, with the total fault pattern recognition success rate almost 99.09% based on a small number of training samples, which shows a certain improvement in diagnostic accuracy compared with the existing intelligent diagnostic methods from [34–36]. The time cost of the proposed method through a laptop computer with a 4.0 GHz dual processor for one test case is only 0.016 s. The time consumption of the proposed approach is encouraging, and the proposed method may be suitable for online bearing fault diagnosis. For supplementary verification, the $k$-fold cross-validation is performed for those 550 data samples and the average success rate is 100% for 10-fold cross-validation, and the average success rate is 99.98% for fivefold cross-validation.

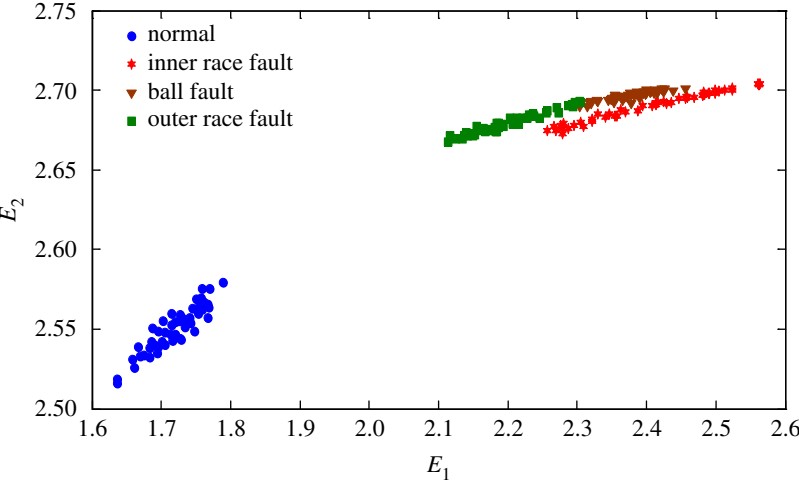

**Figure 4.** Entropy characteristics of a random selected sample from normal operating condition and various fault conditions with fault diameter 7‰, where the abscissa axis $E_1$ represents the Shannon entropy, and the ordinate axis $E_2$ represents the exponential entropy.

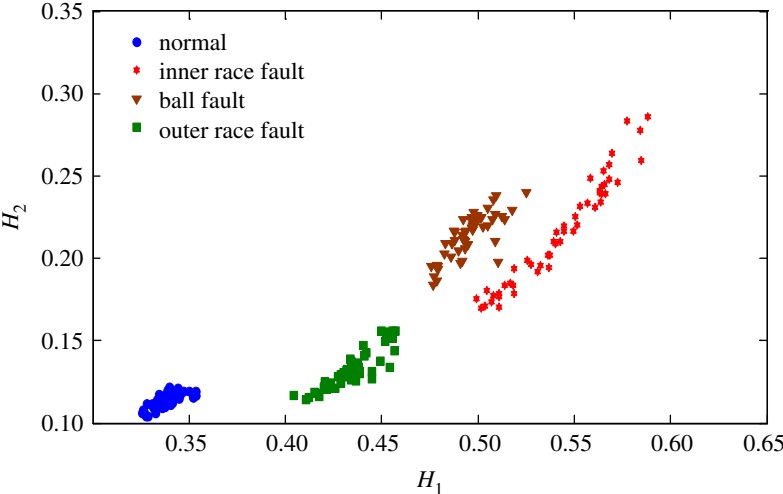

**Figure 5.** Holder coefficient characteristics of a random selected sample from normal operating condition and various fault conditions with fault diameter 7‰, where the abscissa axis $H_1$ represents the Holder coefficient with the rectangular sequence as the reference sequence.

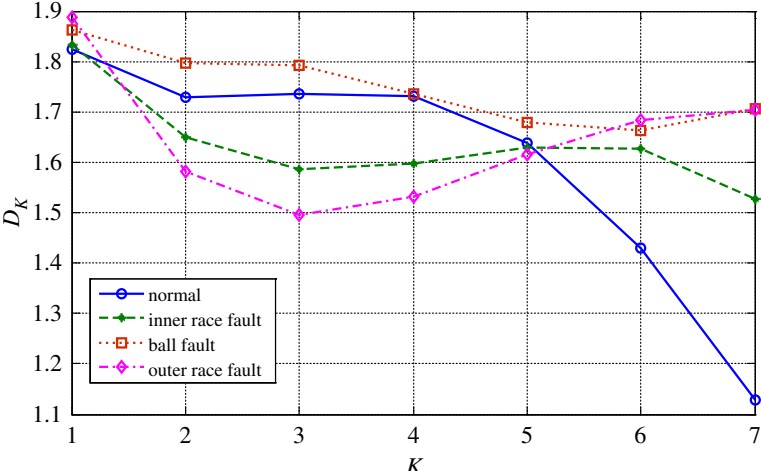

**Figure 6.** Improved generalized box-counting dimension characteristics of a random chosen sample from bearing normal condition and different fault conditions with fault size 7‰.

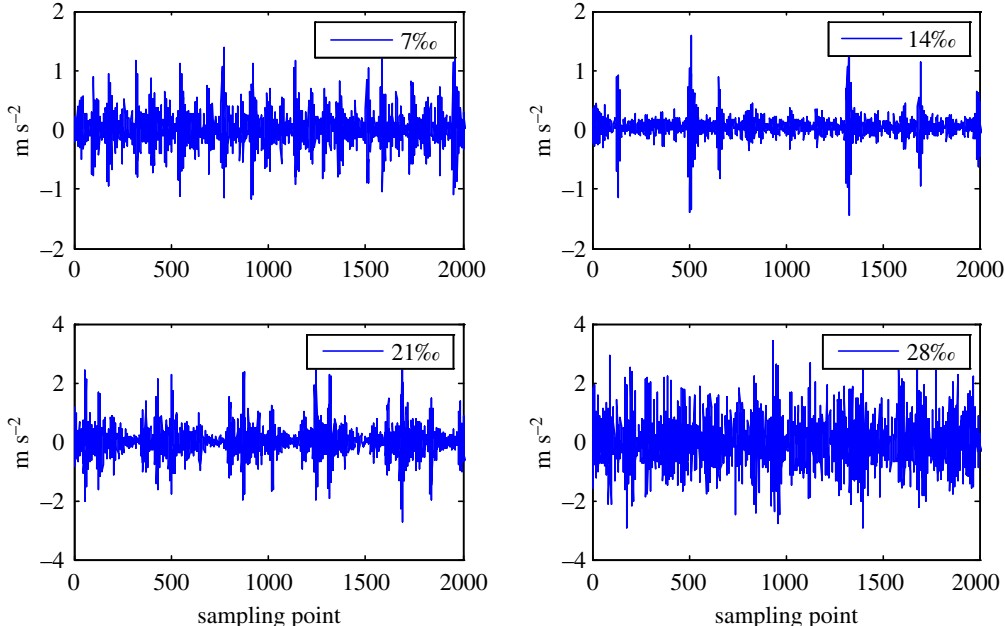

**Figure 7.** Bearing inner race fault conditions with various severities.

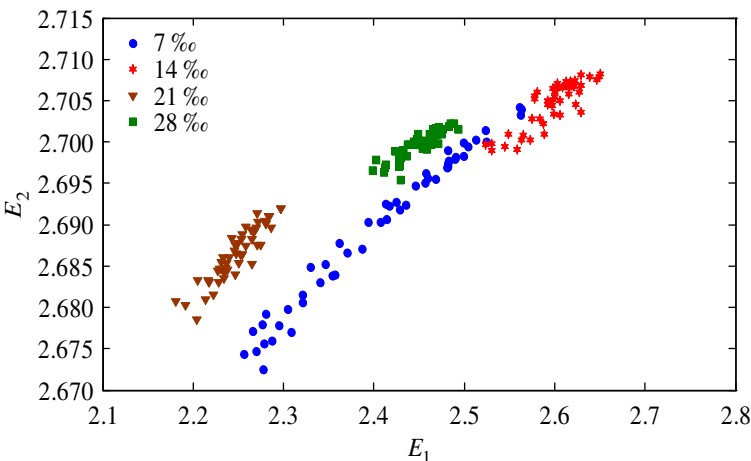

**Figure 8.** Entropy characteristics of a random selected sample from inner race fault condition with various severities.

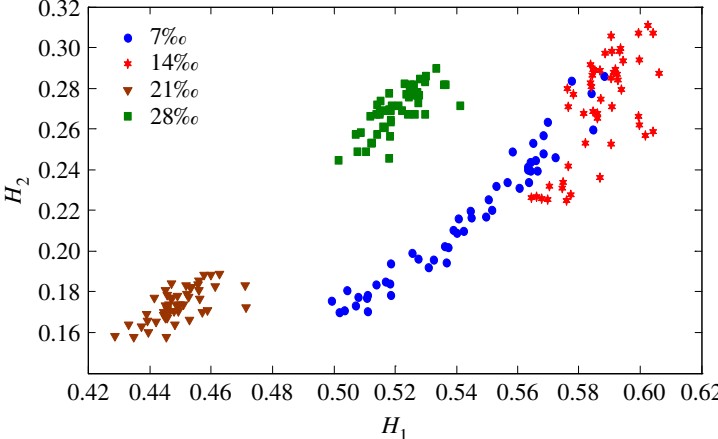

**Figure 9.** Holder coefficient characteristics of a random selected sample from inner race fault condition with various severities.

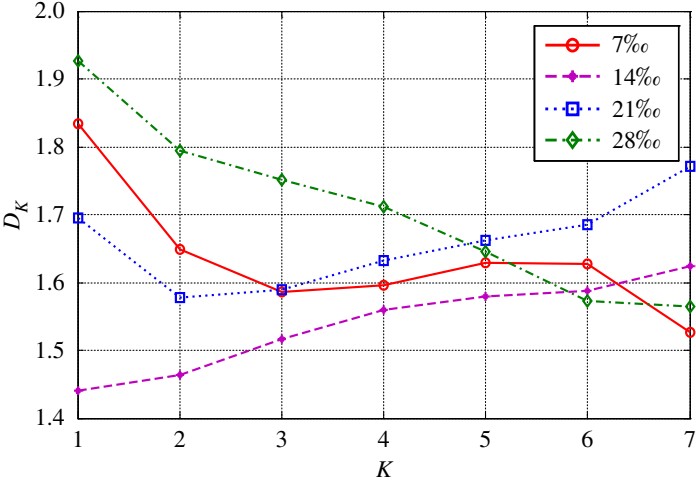

**Figure 10.** Improved generalized box-counting dimension characteristics of a random chosen sample from bearing inner race fault condition with different levels of severity.

**Table 2.** The diagnostic results by the proposed method compared with results from [34–36].

| label of classification | the number of testing samples | the number of misclassified samples | | | | testing accuracy (%) | | | |
|---|---|---|---|---|---|---|---|---|---|
| | | [34] | [35] | [36] | proposed | [34] | [35] | [36] | proposed |
| 1 | 40 | 0 | 0 | 0 | 0 | 100 | 100 | 100 | 100 |
| 2 | 40 | 0 | 0 | 0 | 0 | 100 | 100 | 100 | 100 |
| 3 | 40 | 0 | 4 | 2 | 2 | 100 | 90 | 95 | 95 |
| 4 | 40 | 3 | 0 | 0 | 0 | 92.5 | 100 | 100 | 100 |
| 5 | 40 | 0 | 0 | 0 | 0 | 100 | 100 | 100 | 100 |
| 6 | 40 | 2 | 4 | 3 | 0 | 95 | 90 | 92.5 | 100 |
| 7 | 40 | 3 | 0 | 0 | 2 | 92.5 | 100 | 100 | 95 |
| 8 | 40 | 3 | 4 | 4 | 0 | 92.5 | 90 | 90 | 100 |
| 9 | 40 | 0 | 0 | 0 | 0 | 100 | 100 | 100 | 100 |
| 10 | 40 | 0 | 0 | 3 | 0 | 100 | 100 | 92.5 | 100 |
| 11 | 40 | 4 | 4 | 0 | 0 | 90 | 90 | 100 | 100 |
| in total | 440 | 15 | 16 | 12 | 4 | 96.59 | 96.36 | 96.97 | 99.09 |

# 4. Conclusion

In this paper, a novel diagnostic framework for rolling bearing fault diagnosis is proposed to fulfil the requirements for accurate assessment of different fault types and severities with real-time computational performance. The related experimental study has illustrated the following conclusions: the diagnostic success rate for bearing faulty conditions can reach 100%, with the total diagnostic success rate almost 99.09% based on a small number of training samples; the proposed approach can improve the fault diagnostic accuracy compared with the existing intelligent diagnostic methods, and may be suitable for online bearing fault diagnosis.

In future research, so as to continually improve the fault diagnostic accuracy, the improvement of feature extraction algorithm and evidence fusion theory can be further carried out, on the prerequisite of algorithm real-time performance.

Data accessibility. The rolling bearing vibration signals for testing are from Case Western Reserve University Bearing Data Center [33] in this paper. The Case Western Reserve University Bearing Data Center, http://csegroups.case.edu/bearingdatacenter/pages/download-data-file (accessed 11 October 2017).

Authors' contributions. J.L. and D.B. participated in data analysis, participated in the design of the study and drafted the manuscript; Y.Y. and X.C. proposed the feature extraction algorithm; Y.X. proposed the improved feature extraction algorithm; Y.R. and S.X. participated in the modification of the revised paper. All authors gave final approval for publication.

Competing interests. The authors declare that there are no conflict of interests regarding the publication of this paper.

Funding. The research of this paper is supported by the National Natural Science Foundation of China (grant no. 51806135 and grant no. 61603239) and the State Key Lab of CEMEE Foundation (CEMEE2016K0102A).

Acknowledgements. The authors are grateful to Case Western Reserve University Bearing Data Center for kindly providing the experimental data.

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
