## [Reviewer comments · Royal Society Open Science]

Review History

RSOS-180773.R0 (Original submission)

Review form: Reviewer 1

Is the manuscript scientifically sound in its present form?

No

Are the interpretations and conclusions justified by the results?

No

Is the language acceptable?

No

Is it clear how to access all supporting data?

No

Do you have any ethical concerns with this paper?

No

Have you any concerns about statistical analyses in this paper?

Yes

Recommendation?

Major revision is needed (please make suggestions in comments)

Comments to the Author(s)

The authors aim to propose an online bearing fault diagnosis, however, the content of the paper did not organise in that way. The content of the paper described a typical bearing faults classification by multi-dimensional features.

1. The paper aims to solve the problem that the traditional time and frequency domain methods are not easy to make an accurate assessment of the health status of rolling bearings. However, the authors did not clearly describe the problems of the traditional time and frequency domain methods. Is that difficulty in extracting useful information from a noisy or nonlinear vibration signal? If yes, please compare/cite the renowned researches in this field.
2. Yager algorithm seems to be the main classifier in the bearing faults classification, however, the description of the Yager algorithm is not available. Please explain how Yager algorithm was adapted into this study.
3. Partial data was used to develop the model, and remaining data was used to test the model. Cross-validation such as k-fold cross-validation is recommended to obtain an average accuracy instead of single accuracy value to test the model ability in actual practical environment.
4. The paper was written in a readable English, but it is not free from grammatical or spelling error such as healthy status vs. health status. Thus, linguistic assistance may be required.

Review form: Reviewer 2 (Jiande Wu)**Is the manuscript scientifically sound in its present form?**

Yes

Are the interpretations and conclusions justified by the results?

No

Is the language acceptable?

No

Is it clear how to access all supporting data?

Yes

Do you have any ethical concerns with this paper?

No

Have you any concerns about statistical analyses in this paper?

No

Recommendation?

Reject

Comments to the Author(s)

A hybrid method combining Multi-dimensional feature extraction method and gray relation algorithm (GRA) is proposed to realize rolling bearing fault diagnosis. Actually, these algorithms have been used by previous researchers for denoising, signal decomposition and envelope analysis, respectively, which could be found in available literature. All of the algorithms used in the manuscript are proposed by other researchers. This reviewer could not find any novelty of the proposed algorithm from the manuscript.

Review form: Reviewer 3 (Xiaoyuan Zhang)

Is the manuscript scientifically sound in its present form?

Yes

Are the interpretations and conclusions justified by the results?

Yes

Is the language acceptable?

Yes

Is it clear how to access all supporting data?

No

Do you have any ethical concerns with this paper?

No

Have you any concerns about statistical analyses in this paper?

No

Recommendation?

Accept with minor revision (please list in comments)

Comments to the Author(s)

The authors of this manuscript proposed a novel rolling bearing fault diagnostic method, in which, multi-dimensional features: entropy characteristics, Holder coefficient characteristics and improved generalized box-counting dimension characteristics were extracted to form the knowledge base. And secondly a gray relation algorithm was used to acquire basic belief assignments, and at last the basic belief assignments were fused through Yager algorithm for achieving bearing fault pattern recognition intelligently. This work has some interesting innovations and conclusions. In my opinion, it can be accepted after minor revisions. The specific needs to be revised are as follows:

1 The grammar and some sentences need further modification and improvement.

2 Some sentences are not concise enough.

3 The bearing data from Case Western Reserve University Bearing Data Center is too old and very easy to classify, hence, I suggest a new dataset to illustrate your model.

4 The time cost of the proposed method is 0.016 seconds. It cannot just rely on this to show that your method is suitable for on-line bearing fault diagnosis.

Decision letter (RSOS-180773.R0)

13-Aug-2018

Dear Dr Ying:

Manuscript ID RSOS-180773 entitled "RESEARCH ON ROLLING BEARING ON-LINE FAULT DIAGNOSIS BASED ON MULTI-DIMENSIONAL FEATURE EXTRACTION AND DEMPSTER-SHAFER EVIDENCE THEORY" which you submitted to Royal Society Open Science, has been reviewed. The comments from reviewers are included at the bottom of this letter.

In view of the criticisms of the reviewers, the manuscript has been rejected in its current form. However, a new manuscript may be submitted which takes into consideration these comments.

Please note that resubmitting your manuscript does not guarantee eventual acceptance, and that your resubmission will be subject to peer review before a decision is made.

Your resubmitted manuscript should be submitted by 10-Feb-2019. If you are unable to submit by this date please contact the Editorial Office.

Please note that Royal Society Open Science will introduce article processing charges for all new submissions received from 1 January 2018. Charges will also apply to papers transferred to Royal Society Open Science from other Royal Society Publishing journals, as well as papers submitted as part of our collaboration with the Royal Society of Chemistry (<http://rsos.royalsocietypublishing.org/chemistry>). If your manuscript is submitted and accepted for publication after 1 Jan 2018, you will be asked to pay the article processing charge, unless you request a waiver and this is approved by Royal Society Publishing. You can find out more about the charges at <http://rsos.royalsocietypublishing.org/page/charges>. Should you have any queries, please contact openscience@royalsociety.org.

Kind regards,
Andrew Dunn
Senior Publishing Editor
Royal Society Open Science Editorial Office
Royal Society Open Science
openscience@royalsociety.org

on behalf of Dr Derek Abbott (Associate Editor) and R. Kerry Rowe (Subject Editor)
openscience@royalsociety.org

Reviewers' Comments to Author:

Reviewer: 1

Comments to the Author(s)

The authors aim to propose an online bearing fault diagnosis, however, the content of the paper did not organise in that way. The content of the paper described a typical bearing faults classification by multi-dimensional features.

1. The paper aims to solve the problem that the traditional time and frequency domain methods are not easy to make an accurate assessment of the health status of rolling bearings. However, the authors did not clearly describe the problems of the traditional time and frequency domain methods. Is that difficulty in extracting useful information from a noisy or nonlinear vibration signal? If yes, please compare/cite the renowned researches in this field.
2. Yager algorithm seems to be the main classifier in the bearing faults classification, however, the description of the Yager algorithm is not available. Please explain how Yager algorithm was adapted into this study.
3. Partial data was used to develop the model, and remaining data was used to test the model. Cross-validation such as k-fold cross-validation is recommended to btain an average accuracy instead of signle accuracy value to test the model ability in actual practical environment.
4. The paper was written in a readable English, but it is not free from grammartical or spelling error such as healthy status vs. health status. Thus, linguistic assistance may required.

Reviewer: 2

Comments to the Author(s)

A hybrid method combining Multi-dimensional feature extraction mehtod and gray relation algorithm (GRA) is proposed to realize rolling bearing fault diagnosis. Actually, these algorithms have been used by previous researchers for denoising, signal decomposition and envelope analysis, respectively, which could be found in available literature. All of the algorithms used in the manuscript are proposed by other researchers. This reviewer could not find any novelty of the proposed algorithm from the manuscript.

Reviewer: 3

Comments to the Author(s)

The authors of this manuscript proposed a novel rolling bearing fault diagnostic method, in which, multi-dimensional features: entropy characteristics, Holder coefficient characteristics and improved generalized box-counting dimension characteristics were extracted to form the knowledge base. And secondly a gray relation algorithm was used to acquire basic belief assignments, and at last the basic belief assignments were fused through Yager algorithm for achieving bearing fault pattern recognition intelligently. This work has some interesting innovations and conclusions. In my opinion, it can be accepted after minor revisions. The specific needs to be revised are as follows:

- 1 The grammar and some sentences need further modification and improvement.
- 2 Some sentences are not concise enough.
- 3 The bearing data from Case Western Reserve University Bearing Data Center is too old and very easy to classify, hence, I suggest a new dataset to illustrate your model.
- 4 The time cost of the proposed method is 0.016 seconds. It cannot just rely on this to show that your method is suitable for on-line bearing fault diagnosis.

Author's Response to Decision Letter for (RSOS-180773.R0)

See Appendix A.

RSOS-181488.R0

Review form: Reviewer 1

Is the manuscript scientifically sound in its present form?

Yes

Are the interpretations and conclusions justified by the results?

Yes

Is the language acceptable?

Yes

Is it clear how to access all supporting data?

Yes

Do you have any ethical concerns with this paper?

No

Have you any concerns about statistical analyses in this paper?

No

Recommendation?

Accept as is

Comments to the Author(s)

All previous comments have been addressed by the authors.

Review form: Reviewer 2 (Jiande Wu)

Is the manuscript scientifically sound in its present form?

Yes

Are the interpretations and conclusions justified by the results?

No

Is the language acceptable?

Yes

Is it clear how to access all supporting data?

Yes

Do you have any ethical concerns with this paper?

No

Have you any concerns about statistical analyses in this paper?

No

Recommendation?

Major revision is needed (please make suggestions in comments)

Comments to the Author(s)

In the revised manuscript, there seems to be no response to the questions raised before. The following questions are outstanding.

1. The authors aim to propose an online bearing fault diagnosis. However, the content of the paper did not organise in that way.
2. According to the extracted features, the good classification results can be achieved by using Holder coefficients. As shown in Figure 5, Holder coefficients are completely separable under different operating conditions.
3. From the characteristics of different fault level of severity, the fusion features seem indistinguishable. Because there is a common overlap and confusion area between the 3 characteristics.

Decision letter (RSOS-181488.R0)

04-Dec-2018

Dear Dr Ying,

The Subject Editor assigned to your paper ("Research on rolling bearing fault diagnosis based on multi-dimensional feature extraction and evidence fusion theory") has now received comments from reviewers. We would like you to revise your paper in accordance with the referee and Associate Editor suggestions which can be found below (not including confidential reports to the Editor). Please note this decision does not guarantee eventual acceptance.

Please ensure you fully respond to the concerns of the reviewers. If you do not satisfy them that the paper is ready for publication in the revision, your manuscript may be rejected from further consideration.

Please submit a copy of your revised paper before 27-Dec-2018. Please note that the revision deadline will expire at 00.00am on this date. If we do not hear from you within this time then it will be assumed that the paper has been withdrawn. In exceptional circumstances, extensions may be possible if agreed with the Editorial Office in advance. We do not allow multiple rounds of revision so we urge you to make every effort to fully address all of the comments at this stage. If deemed necessary by the Editors, your manuscript will be sent back to one or more of the original reviewers for assessment. If the original reviewers are not available we may invite new reviewers.

To revise your manuscript, log into <http://mc.manuscriptcentral.com/rsos> and enter your Author Centre, where you will find your manuscript title listed under "Manuscripts with Decisions." Under "Actions," click on "Create a Revision." Your manuscript number has been

appended to denote a revision. Revise your manuscript and upload a new version through your Author Centre.

When submitting your revised manuscript, you must respond to the comments made by the referees and upload a file "Response to Referees" in "Section 6 - File Upload". Please use this to document how you have responded to each of the comments, and the adjustments you have made. In order to expedite the processing of the revised manuscript, please be as specific as possible in your response.

- Ethics statement

- Data accessibility

If you wish to submit your supporting data or code to Dryad (<http://datadryad.org/>), or modify your current submission to dryad, please use the following link:
<http://datadryad.org/submit?journalID=RSOS&manu=RSOS-181488>

- Competing interests

- Authors' contributions

AB carried out the molecular lab work, participated in data analysis, carried out sequence alignments, participated in the design of the study and drafted the manuscript; CD carried out the statistical analyses; EF collected field data; GH conceived of the study, designed the study,

coordinated the study and helped draft the manuscript. All authors gave final approval for publication.

- Acknowledgements

- Funding statement

Please note that Royal Society Open Science charge article processing charges for all new submissions that are accepted for publication. Charges will also apply to papers transferred to Royal Society Open Science from other Royal Society Publishing journals, as well as papers submitted as part of our collaboration with the Royal Society of Chemistry (<http://rsos.royalsocietypublishing.org/chemistry>). If your manuscript is newly submitted and subsequently accepted for publication, you will be asked to pay the article processing charge, unless you request a waiver and this is approved by Royal Society Publishing. You can find out more about the charges at <http://rsos.royalsocietypublishing.org/page/charges>. Should you have any queries, please contact openscience@royalsociety.org.

on behalf of Dr Derek Abbott (Associate Editor) and R. Kerry Rowe (Subject Editor)
openscience@royalsociety.org

Reviewer comments to Author:

Reviewer: 2

Comments to the Author(s)

In the revised manuscript, there seems to be no response to the questions raised before. The following questions are outstanding.

1. The authors aim to propose an online bearing fault diagnosis. However, the content of the paper did not organise in that way.
2. According to the extracted features, the good classification results can be achieved by using Holder coefficients. As shown in Figure 5, Holder coefficients are completely separable under different operating conditions.
3. From the characteristics of different fault level of severity, the fusion features seem indistinguishable. Because there is a common overlap and confusion area between the 3 characteristics.

Reviewer: 1

Comments to the Author(s)

All previous comments have been addressed by the authors.

Author's Response to Decision Letter for (RSOS-181488.R0)

See Appendix B.

Decision letter (RSOS-181488.R1)

15-Jan-2019

Dear Dr Ying,

I am pleased to inform you that your manuscript entitled "Research on rolling bearing fault diagnosis based on multi-dimensional feature extraction and evidence fusion theory" is now accepted for publication in Royal Society Open Science.

on behalf of Dr Derek Abbott (Associate Editor) and Professor R. Kerry Rowe (Subject Editor)
openscience@royalsociety.org

Appendix A

Journal title: Royal Society Open Science

Manuscript title: Research on rolling bearing fault diagnosis based on multi-dimensional feature extraction and evidence fusion theory

Dear editor,

Thank you for your useful comments and suggestions on our manuscript. We have modified the manuscript accordingly, and the detailed corrections are listed below:

Reviewers' Comments to Author:

Reviewer: 1

Comments to the Author(s)

The authors aim to propose an online bearing fault diagnosis, however, the content of the paper did not organise in that way. The content of the paper described a typical bearing faults classification by multi-dimensional features.

1. The paper aims to solve the problem that the traditional time and frequency domain methods are not easy to make an accurate assessment of the health status of rolling bearings. However, the authors did not clearly describe the problems of the traditional time and frequency domain methods. Is that difficulty in extracting useful information from a noisy or nonlinear vibration signal? If yes, please compare/cite the renowned researches in this field.

Answer: The background introduction of the paper has been modified, thanks for your advice!

2. Yager algorithm seems to be the main classifier in the bearing faults classification, however, the description of the Yager algorithm is not available. Please explain how Yager algorithm was adapted into this study.

Answer: The description of the Yager algorithm has been added in the paper, thanks for your advice!

3. Partial data was used to develop the model, and remaining data was used to test the model. Cross-validation such as k-fold cross-validation is recommended to obtain an average accuracy instead of single accuracy value to test the model ability in actual practical environment.

Answer: The modification has been made in the paper.

The diagnostic results from Table 2 show that the detecting success rate for bearing faulty conditions can reach 100%, with the total fault pattern recognition success rate almost 99.09% **based on a small number of training samples**, which shows a certain improvement in diagnostic accuracy compared with the **existing intelligent diagnostic methods** from references [33], [34] and [35]. The time cost of the proposed method through a laptop computer with a 4.0 GHz dual processor for one Test Case is only 0.016 seconds. The time consumption of the proposed approach is encouraging, and the proposed method **may be** suitable for on-line bearing fault

diagnosis. For supplementary verification, the k -fold cross validation is performed for those 550 data samples and the average success rate is 100% for 10-fold cross validation, and the average success rate is 99.98% for 5-fold cross validation.

4. The paper was written in a readable English, but it is not free from grammatical or spelling error such as healthy status vs. health status. Thus, linguistic assistance may be required.

Answer: The linguistic modification has been made in the paper. Thanks for your advice!

Reviewer: 2

Comments to the Author(s)

A hybrid method combining Multi-dimensional feature extraction method and gray relation algorithm (GRA) is proposed to realize rolling bearing fault diagnosis. Actually, these algorithms have been used by previous researchers for denoising, signal decomposition and envelope analysis, respectively, which could be found in available literature. All of the algorithms used in the manuscript are proposed by other researchers. This reviewer could not find any novelty of the proposed algorithm from the manuscript.

Answer: With the development of artificial intelligence, the procedure of rolling bearing fault diagnosis is more and more treated as a procedure of pattern recognition, and its effectiveness and reliability mainly depend on the selection of dominant characteristic vector of the fault features. In the paper, a novel **diagnostic framework** for rolling bearing faults based on multi-dimensional feature extraction and evidence fusion theory, is proposed to fulfill the requirements for effective assessment of different fault types and severities with real-time computational performance.

Fig.1 The **diagnostic framework for rolling bearing fault diagnosis**

The diagnostic results from Table 2 show that the detecting success rate for bearing faulty conditions can reach 100%, with the total fault pattern recognition success rate almost 99.09% **based on a small number of training samples**, which shows a certain improvement in diagnostic accuracy compared with the **existing intelligent diagnostic methods** from references [33], [34] and [35]. The time cost of the proposed method through a laptop computer with a 4.0 GHz dual processor for one Test Case is only 0.016 seconds. The time consumption of the proposed approach is encouraging, and the proposed method **may be** suitable for on-line bearing fault diagnosis. For supplementary verification, the k -fold cross validation is performed for those 550 data samples and the average success rate is 100% for 10-fold cross validation, and the average

success rate is 99.98% for 5-fold cross validation.

Reviewer: 3

Comments to the Author(s)

The authors of this manuscript proposed a novel rolling bearing fault diagnostic method, in which, multi-dimensional features: entropy characteristics, Holder coefficient characteristics and improved generalized box-counting dimension characteristics were extracted to form the knowledge base. And secondly a gray relation algorithm was used to acquire basic belief assignments, and at last the basic belief assignments were fused through Yager algorithm for achieving bearing fault pattern recognition intelligently. This work has some interesting innovations and conclusions. In my opinion, it can be accepted after minor revisions. The specific needs to be revised are as follows:

1 The grammar and some sentences need further modification and improvement.

Answer: The linguistic modification has been made in the paper. Thanks for your advice!

2 Some sentences are not concise enough.

Answer: The related modification has been made in the paper. Thanks for your advice!

3 The bearing data from Case Western Reserve University Bearing Data Center is too old and very easy to classify, hence, I suggest a new dataset to illustrate your model.

Answer: This bearing database from Case Western Reserve University Bearing Data Center is a classical database, which has been widely used for testing the effectiveness of the proposed diagnostic methods by researchers, seen in "Reference". And in order to comparison with the existing intelligent diagnostic methods, we used this bearing database to test our proposed method. Thanks for your advice!

4 The time cost of the proposed method is 0.016 seconds. It cannot just rely on this to show that your method is suitable for on-line bearing fault diagnosis.

Answer: The expression has been modified as "may be suitable for online bearing fault diagnosis."

Appendix B

Journal title: Royal Society Open Science

Manuscript title: Research on rolling bearing fault diagnosis based on multi-dimensional feature extraction and evidence fusion theory

Dear editor,

Thank you for your useful comments and suggestions on our manuscript. We have modified the manuscript accordingly, and the detailed corrections are listed below:

Reviewer(s)' Comments to Author:

Reviewer comments to Author:

Reviewer: 2

Comments to the Author(s)

In the revised manuscript, there seems to be no response to the questions raised before. The following questions are outstanding.

1. The authors aim to propose an online bearing fault diagnosis. However, the content of the paper did not organise in that way.

Answer: We have modified our paper title from “Research on rolling bearing on-line fault diagnosis based on multi-dimensional feature extraction and Dempster-shafer Evidence theory” into “Research on rolling bearing fault diagnosis based on multi-dimensional feature extraction and evidence fusion theory” so as to fit for the content of the paper.

2. According to the extracted features, the good classification results can be achieved by using Holder coefficients. As shown in Figure 5, Holder coefficients are completely separable under different operating conditions.

Answer: From Fig.4, Fig.5 and Fig.6, the Holder coefficient characteristics show a better inter-class separation and intra-class polymerization than entropy characteristics and improved generalized box-counting dimension characteristics extracted from the bearing vibration signals with different fault types. We have modified this part in paper.

3. From the characteristics of different fault level of severity, the fusion features seem indistinguishable. Because there is a common overlap and confusion area between the 3 characteristics.

Answer: From Fig.4, Fig.5 and Fig.6, the Holder coefficient characteristics show a better inter-class separation and intra-class polymerization than entropy characteristics and improved generalized box-counting dimension characteristics extracted from the bearing vibration signals with different fault types. However, from Fig.8, Fig.9 and Fig.10, the improved generalized box-counting dimension characteristics show a better inter-class separation than entropy characteristics and Holder coefficient characteristics extracted from the bearing vibration signals with different severities. As entropy characteristics, Holder coefficient characteristics and improved generalized box-counting dimension characteristics show their strengths and weaknesses in classifying

different fault types and severities, evidence fusion theory is used to obtain the final diagnostic results (i.e., fault types and severities) after the fusion of BBAs. We have modified this part in paper.

Thanks a lot for your advise!

Reviewer: 1

Comments to the Author(s)

All previous comments have been addressed by the authors.

Answer: Thanks a lot for your advise!